# Is Long Horizon RL More Difficult Than Short Horizon RL?

**Ruosong Wang**[*]
Carnegie Mellon University
ruosongw@andrew.cmu.edu

**Simon S. Du**[*]
University of Washington, Seattle
ssdu@cs.washington.edu

**Lin F. Yang**[*]
University of California, Los Angles
linyang@ee.ucla.edu

**Sham M. Kakakde**
University of Washington, Seattle and Microsoft Research
sham@cs.washington.edu

## Abstract

Learning to plan for long horizons is a central challenge in episodic reinforcement learning problems. A fundamental question is to understand how the difficulty of the problem scales as the horizon increases. Here the natural measure of sample complexity is a normalized one: we are interested in the *number of episodes* it takes to provably discover a policy whose value is $\varepsilon$ near to that of the optimal value, where the value is measured by the *normalized* cumulative reward in each episode. In a COLT 2018 open problem, Jiang and Agarwal conjectured that, for tabular, episodic reinforcement learning problems, there exists a sample complexity lower bound which exhibits a polynomial dependence on the horizon — a conjecture which is consistent with all known sample complexity upper bounds. This work refutes this conjecture, proving that tabular, episodic reinforcement learning is possible with a sample complexity that scales only *logarithmically* with the planning horizon. In other words, when the values are appropriately normalized (to lie in the unit interval), this results shows that long horizon RL is no more difficult than short horizon RL, at least in a minimax sense.

Our analysis introduces two ideas: (i) the construction of an $\varepsilon$-net for near-optimal policies whose log-covering number scales only logarithmically with the planning horizon, and (ii) the Online Trajectory Synthesis algorithm, which adaptively evaluates all policies in a given policy class and enjoys a sample complexity that scales logarithmically with the cardinality of the given policy class. Both may be of independent interest.

## 1 Introduction

Long horizons, along with the state dependent transitions, is the differentiator between reinforcement learning (RL) problems and simpler contextual bandit problems. In the former (RL), actions taken at early stages could substantially impact the future; with regards to planning, the agent must not only consider the immediate reward but also the possible future transitions into differing states. In contrast, for the latter (contextual bandit problems), the action taken at each time step is independent

---

[*]Equal contribution

of the future, though it does depend on the current state (the "context"); we can consider a contextual bandit problem as a Markov decision process (MDP) with a horizon equal to one. For a known contextual bandit problem, it is sufficient for the agent to act myopically by choosing the action which maximizes the current reward as a function of the current state.

Jiang and Agarwal [2018] proposed to study this distinction by examining how the sample complexity depends on the horizon length (of each episode) in a finite horizon, episodic MDP, where the MDP is unknown to the agent. Clearly, as the horizon $H$ grows, we will observe more samples in each episode. To appropriately measure the sample complexity (see [Jiang and Agarwal, 2018]), we consider a normalized notion: we are interested in the *number of episodes* it takes to provably discover a policy whose value is $\varepsilon$ near to that of the optimal value, where the value is measured by the *normalized* cumulative reward in each episode (i.e. values are normalized to be bounded between 0 and 1). Here, all existing upper bounds depend polynomially on the horizon, while lower bounds do not provide *any* dependence on the horizon $H$. Motivated by these observations, Jiang and Agarwal [2018] posed the following open problem in COLT 2018:

*Can we prove a lower bound that depends polynomially on the planning horizon, $H$?*

Jiang and Agarwal [2018] conjectured a linear dependence on the horizon, which is consistent with all existing upper bounds, which scale at least linearly with the planning horizon [Dann and Brunskill, 2015, Azar et al., 2017, Zanette and Brunskill, 2019] (see Section 3 for further discussion). In other words, the conjecture is that, even when the values are appropriately normalized, long horizon RL is polynomially more difficult than short horizon RL.

This work resolves this question, with, perhaps surprisingly, a *negative* answer. Here we give an informal version of our main result.

**Theorem 1.1** (Informal version of Theorem 4.1). *Suppose the reward at each time step is non-negative and the total reward of each episode is upper bounded by 1. Given any target accuracy $0 < \varepsilon < 1$ and a failure probability $0 \le \delta \le 1$, the* Online Trajectory Synthesis *algorithm returns an $\varepsilon$-optimal policy with probability at least $1 - \delta$ by sampling at most* poly $(|\mathcal{S}|, |\mathcal{A}|, \log H, 1/\varepsilon, \log(1/\delta))$ *episodes, where $|\mathcal{S}|$ is the number of states and $|\mathcal{A}|$ is the number actions.*

Importantly, this sample complexity scales only *logarithmically* with $H$. Thus, there does not exist a lower bound that depends polynomially on the planning horizon. This result is an exponential improvement on the dependency on $H$ over existing upper bounds.[2]

In the context of the discussion in [Jiang and Agarwal, 2018], these results suggest that perceived differences between long horizon RL and contextual bandit problems (or short horizon RL) are not attributable to the horizon dependence, at least in a minimax sense. It is worthwhile to note that while our upper bound is logarithmic in $H$, it does have polynomial dependence (beyond just being linear) on the number of states (or "contexts") and the number actions. We return to the question of obtaining an optimal rate in Section 6.

## 2 Preliminaries

Throughout this paper, for a given integer $H$, we use $[H]$ to denote the set $\{1, 2, \ldots, H\}$. For a condition $\mathcal{E}$, we use $\mathbb{I}[\mathcal{E}]$ to denote the indicator function, i.e., $\mathbb{I}[\mathcal{E}] = 1$ if $\mathcal{E}$ holds and $\mathbb{I}[\mathcal{E}] = 0$ otherwise.

Let $M = (\mathcal{S}, \mathcal{A}, P, R, H, \mu)$ be a *Markov Decision Process* (MDP) where $\mathcal{S}$ is the finite state space, $\mathcal{A}$ is the finite action space, $P : \mathcal{S} \times \mathcal{A} \to \Delta(\mathcal{S})$ is the transition operator which takes a state-action pair and returns a distribution over states, $R : \mathcal{S} \times \mathcal{A} \to \Delta(\mathbb{R})$ is the reward distribution, $H \in \mathbb{Z}_+$ is the planning horizon (episode length), and $\mu \in \Delta(\mathcal{S})$ is the initial state distribution. We refer to a *contextual bandit problem* as an MDP with $H = 1$.

A (non-stationary) policy $\pi$ chooses an action $a$ based on the current state $s \in \mathcal{S}$ and the time step $h \in [H]$. Formally, $\pi = \{\pi_h\}_{h=1}^H$ where for each $h \in [H]$, $\pi_h : \mathcal{S} \to \mathcal{A}$ maps a given state to

an action. The policy $\pi$ induces a (random) trajectory $s_1, a_1, r_1, s_2, a_2, r_2, \ldots, s_H, a_H, r_H$, where $s_1 \sim \mu$, $a_1 = \pi_1(s_1)$, $r_1 \sim R(s_1, a_1)$, $s_2 \sim P(s_1, a_1)$, $a_2 = \pi_2(s_2)$, etc.

We assume, almost surely, that $r_h \geq 0$ for all $h \in [H]$ and

$$\sum_{h=1}^{H} r_h \in [0, 1].$$

In other words, we work with the normalized cumulative reward. It is worth emphasizing that this assumption is weaker than the standard one in that we do not assume the immediate rewards $r_h$ are bounded (see Assumptions 3.1 and 3.2 in Section 3 for comparison). Our goal is to find a policy $\pi$ that maximizes the expected total reward, i.e.

$$\max_{\pi} \mathbb{E}\left[\sum_{h=1}^{H} r_h \mid \pi\right].$$

We say a policy $\pi$ is $\varepsilon$-optimal if $\mathbb{E}\left[\sum_{h=1}^{H} r_h \mid \pi\right] \geq \mathbb{E}\left[\sum_{h=1}^{H} r_h \mid \pi^*\right] - \varepsilon$, where $\pi^*$ denotes an optimal policy.

An important concept in RL is the $Q$-function. Given a policy $\pi$, a level $h \in [H]$ and a state-action pair $(s, a) \in \mathcal{S} \times \mathcal{A}$, the $Q$-function is defined as:

$$Q_h^\pi(s, a) = \mathbb{E}\left[\sum_{h'=h}^{H} r_{h'} \mid s_h = s, a_h = a, \pi\right].$$

Similarly, the value function of a given state $s \in \mathcal{S}$ is defined as:

$$V_h^\pi(s) = \mathbb{E}\left[\sum_{h'=h}^{H} r_{h'} \mid s_h = s, \pi\right].$$

For notational convenience, we denote $Q_h^*(s, a) = Q_h^{\pi^*}(s, a)$ and $V_h^*(s) = V_h^{\pi^*}(s)$.

## 3 Related Work

We now discuss related theoretical work on tabular RL, largely focusing on the episodic, finite horizon settings due to these being most relevant in our setting (also see [Jiang and Agarwal, 2018]). Here, most results focus either on the regret minimization problem, which aims to collect maximum reward for a limit number of interactions with the environment, or on the sample complexity of PAC learning for finding a near-optimal policy.

In episodic tabular RL, sample complexities will depend on $|\mathcal{S}|$, $|\mathcal{A}|$ and $H$, all of which are assumed to be finite. In many works, the standard assumption on the rewards are that: $r_h \in [0, 1]$ and hence $\sum_{h=1}^{H} r_h \in [0, H]$. However, as pointed out in [Jiang and Agarwal, 2018], to have a fair comparison with contextual bandits (and short horizon RL) and illustrate the hardness due to the planning horizon, one should scale down by an $H$ factor in order to normalize the total reward, so that it is bounded in $[0, 1]$.

Following [Jiang and Agarwal, 2018] (also see [Kakade, 2003]), we can rescale the total reward (the value functions) to be bounded in $[0, 1]$ as follows:

**Assumption 3.1** (Reward Uniformity, the standard assumption). *The reward received at the $h$-th time step $r_h$ satisfies $r_h \in [0, 1/H]$ for $h = 1, \ldots, H$, and hence $\sum_{h=1}^{H} r_h \leq 1$.*

Furthermore, following [Jiang and Agarwal, 2018], we can further relax this assumption to a weaker version where we only bound the total reward as follow.

**Assumption 3.2** (Bounded Total Reward e.g. [Krishnamurthy et al., 2016]). *The reward received at the $h$-th time step $r_h$ satisfies $r_h \geq 0$ for $h = 1, \ldots, H$, and we assume that total reward is bounded as $\sum_{h=1}^{H} r_h \leq 1$.*

Assumption 3.2 is more natural in environments with sparse rewards, as argued in [Jiang and Agarwal, 2018]. As pointed out in [Jiang and Agarwal, 2018], this scaling permits more fair comparisons with contextual bandit problems. Furthermore, Assumption 3.2 is clearly more general than Assumption 3.1, so any bound under Assumption 3.2, also implies a bound under Assumption 3.1.

Under Assumption 3.1, a line of work has attempted to provide tight sample complexity bounds [Azar et al., 2017, Dann and Brunskill, 2015, Dann et al., 2017, 2019, Jin et al., 2018, Osband and Van Roy, 2017]. To obtain an $\varepsilon$-optimal policy, state-of-the-art results show that $\widetilde{O}\left(\frac{|\mathcal{S}||\mathcal{A}|}{\varepsilon^2} + \frac{\text{poly}(|\mathcal{S}|,|\mathcal{A}|,H)}{\varepsilon}\right)$ episodes suffice [Dann et al., 2019, Azar et al., 2017].[3] In particular, the first term matches the lower bound $\Omega\left(\frac{|\mathcal{S}||\mathcal{A}|}{\varepsilon^2}\right)$ up to logarithmic factors [Dann and Brunskill, 2015, Osband and Roy, 2016, Azar et al., 2017].

There are two concerns with these work. First, these bound are optimal only in the regime $\varepsilon \in [0, 1/H]$.[4] However, as explained in [Jiang and Agarwal, 2018], in many scenarios with a long planning horizon such as control, this regime is not interesting.[5] In particular, the more interesting regime is when $\varepsilon \gg 1/H$. Secondly, Assumption 3.1 is a strong assumption as it cannot model environments with sparse rewards. Jiang and Agarwal [2018] thus argued that one should study tabular RL with a more general assumption, i.e. under Assumption 3.2.

Note that environments under Assumption 3.2 can have one-step reward as high as a constant. These are the sparse reward environments which are often considered to be hard. Jiang and Agarwal [2018] conjectured that under Assumption 3.2, there is a lower bound on the sample complexity that scales *polynomially* in $H$. Under Assumption 3.2, upper bounds in [Azar et al., 2017, Dann et al., 2019] will become $\widetilde{O}\left(\frac{|\mathcal{S}||\mathcal{A}|H^2}{\varepsilon^2}\right)$ whose dependency on $H$ is not tight even in the $\varepsilon \ll 1/H$ regime (recall the lower bound is still $\Omega\left(\frac{|\mathcal{S}||\mathcal{A}|}{\varepsilon^2}\right)$). Recently, under Assumption 3.2, Zanette and Brunskill [2019] gave a new algorithm which enjoys a sample complexity of $\widetilde{O}\left(|\mathcal{S}||\mathcal{A}|/\varepsilon^2 + \text{poly}\left(|\mathcal{S}|,|\mathcal{A}|,H\right)/\varepsilon\right)$.[6] Unfortunately, the second term still scales *polynomially* with $H$.

In another related "generative model" setting, where the agent can query samples freely from any state-action pair of the environment, the question of sample complexity is also posed as the total number of batches of queries (a batch corresponds to $H$ queries) to the environment to obtain an $\varepsilon$-optimal policy. Results in this setting include [Kearns and Singh, 1999, Kakade, 2003, Singh and Yee, 1994, Azar et al., 2013, Sidford et al., 2018b,a, Agarwal et al., 2019]. However, even with this much stronger query model, we are not aware of any algorithm whose sample complexity scales sublinearly in $H$.

The main barrier of achieving logarithmic dependency on $H$ is that almost all the above mentioned works rely on a dynamic programming step (i.e., the Bellman update) to learn the optimal value functions. In this paper, we bypass this long-standing barrier using a completely different approach. We provide a new technique to simulate the Monte Carlo methods to evaluate a set of given policies using existing samples and to do exploration. Interestingly, this technique is related to the method proposed in [Fonteneau et al., 2013] for policy evaluation, though there is no exploration component in their paper. Our algorithm has some similarities to the "Trajectory Tree" method in [Kearns et al., 2000] in that both attempt to simultaneously evaluate many policies on a "tree" of collected data (using a generative model); a key difference in our approach is that (due to the adaptive nature of data collection) we are not able to explicitly build a tree or reuse data on trajectories.

## 4 Main Result and Technical Overview

Now we present the main result of this work.

**Theorem 4.1.** *Suppose the reward at each level satisfies $r_h \geq 0$ and $\sum_{h=1}^{H} r_h \leq 1$ almost surely. Given a target accuracy $0 < \varepsilon < 1$, then with probability at least $1 - \delta$, Algorithm 3 returns an $\varepsilon$-optimal policy by sampling at most*

$$O\left(|\mathcal{S}|^3|\mathcal{A}|^3 \log^2 H/\varepsilon^3 \log(|\mathcal{S}||\mathcal{A}|/\varepsilon) \cdot \left(|\mathcal{S}|^2|\mathcal{A}| \log(H|\mathcal{S}|/\varepsilon) + \log(1/\delta)\right)\right)$$

*episodes.*

This result shows that only *logarithmic* dependence on the horizon is possible. Algorithm 3 is provided in Section 5. We remark that, while our bound improves the dependency on $H$, the dependency on $|\mathcal{S}|$, $|\mathcal{A}|$ and $1/\varepsilon$ are worse than existing state-of-the-art bounds (cf. Section 3). It is an open problem to further tighten the dependencies on $|\mathcal{S}|$, $|\mathcal{A}|$ and $1/\varepsilon$.

We now provide an overview of our analysis.

### 4.1 Technical Overview

**An $\varepsilon$-net For Non-stationary Policies.** We first construct a set of polices $\Pi$ which contains an $\varepsilon$-optimal policy for any MDP. Importantly, the size of $\Pi$ satisfies $|\Pi| = (H/\varepsilon)^{\text{poly}(|\mathcal{S}||\mathcal{A}|)}$, which is acceptable since the overall sample complexity of our algorithm depends only logarithmically on $|\Pi|$. To define such a set of policies, we consider all discretized MDPs whose transition probabilities and reward values are integer multiples of $\text{poly}(\varepsilon/(|\mathcal{S}||\mathcal{A}|H))$. Clearly, there are most $(H/\varepsilon)^{\text{poly}(|\mathcal{S}||\mathcal{A}|)}$ such discretized MDPs, and for each discretized MDP $M$, we add an optimal policy of $M$ into $\Pi$. It remains to show that for any $M$, there exists a policy $\pi \in \Pi$ which is an $\varepsilon$-optimal policy of $M$. This can be seen since there exists a discretized MDP $\hat{M}$ whose transition probabilities and reward values are close enough to those of $M$, and by standard perturbation analysis, it can be easily shown that an optimal policy of $\hat{M}$ is an $\varepsilon$-optimal policy of $M$. The formal analysis is given in Section 5.1.

**The Trajectory Synthesis Method.** Now we show how to evaluate values of all policies in the policy set $\Pi$ constructed above by sampling at most $\text{poly}(|\mathcal{S}|, |\mathcal{A}|, 1/\varepsilon, \log |\Pi|, \log H)$ episodes. To achieve this goal, we design a trajectory simulator, which, for every policy in the set, either interacts with the environment to collect trajectories, or simulates trajectories using collected samples. In either case, the simulator obtains trajectories of the policy with distribution close enough to those sampled by interacting with the environment. The most natural idea is to collect trajectories for each policy $\pi$ separately by interacting with the environment. This method, although is guaranteed to output "true" trajectories for every policy, has sample complexity at least linear in the size of the policy set $|\Pi|$ and is thus insufficient for our goal. Another possible way to evaluate policies is to build an empirical model (an estimation of transition probability and reward function) and evaluate policies on the empirical model (or to build a trajectory tree as in Kearns et al. [2000]). However, we do not know how to deal with the dependency issue in building the empirical model and to prove a sample complexity bound that scales logarithmically with the planning horizon. The analysis based on performance difference lemma can lead to polynomial dependency on the planning horizon [Kakade, 2003].

**Reuse Samples.** A key observation is that once we obtain a trajectory for a policy by interacting with the environment, samples collected during this process can be used to simulate trajectories for other policies. To better illustrate this idea, we use $\Pi_{\mathcal{D}}$ to denote the set of policies for which we have obtained trajectories by interacting with the environment, and denote

$$\mathcal{D}_{s,a} = \left[\left(s_{(s,a)}^{(1)}, r_{(s,a)}^{(1)}\right), \left(s_{(s,a)}^{(2)}, r_{(s,a)}^{(2)}\right), \ldots\right]$$

to be the sequence of samples obtained from $P(s,a)$ and $R(s,a)$. These samples are sorted in chronological order. Suppose that now we are given a new policy $\pi$ and for all $(s,a) \in \mathcal{S} \times \mathcal{A}$, $|\mathcal{D}_{s,a}^{(t)}| \geq H$. Then it is easy to simulate a trajectory for $\pi$ using the set of samples $\{\mathcal{D}_{s,a}\}_{s \in \mathcal{S}, a \in \mathcal{A}}$. Indeed, we start from state $s_1$ and set $(s_2, r_2)$ to be the first pair in $\mathcal{D}_{s_1, \pi_1(s_1)}$, and then set $(s_3, r_3)$ to be the first pair in $\mathcal{D}_{s_2, \pi_2(s_2)}$ that has not been used, etc. In general, suppose we are at state $s_h$ for some $h < H$, we set $(s_{h+1}, r_{h+1})$ to be the first pair in $\mathcal{D}_{s_h, \pi_h(s_h)}$ that has not been used. Note that such a procedure generates a trajectory for $\pi$ with exactly the same distribution as that generated by interacting with the environment.

**Avoid Unnecessary Sampling.** We have described the approach to reuse samples in the above paragraph. Nevertheless, there is a problem intrinsic to the above approach: if the process of simulating a policy $\pi$ fails (i.e., some $(s_h, \pi_h(s))$ has been visited $j \le H$ times but $|\mathcal{D}_{s_h, \pi_h(s)}| < j$), should we interact with the environment to generate a trajectory or simply claim failure? Note that claiming failure is acceptable as long as the overall failure probability is small.

In order to decide when to interact with the environment, we design a procedure to estimate the probability of simulation failure. If the failure probability is already small enough, there is no need to interact with the environment. Otherwise, we interact with the environment to obtain a trajectory. To bound the overall sample complexity, one key observation is that if the failure probability is large, then the policy will visit some state-action pair more frequently than all existing policies. In the formal analysis, we make this intuition rigorous by designing a potential function to measure the overall progress made by our algorithm.

## 5 The Algorithm

In this section, for the sake of presentation, we assume a fixed initial state $s_1$. When the initial state is sampled from a distribution $\mu$, we may create a new state $s_0$ and set $s_0$ to be the initial state. We set $P(s_0, a) = \mu$ and $r(s_0, a) = 0$ for all $a \in \mathcal{A}$, and increase the planning horizon $H$ by 1. By doing so, now $s_1$ is sampled from the initial state distribution $\mu$.

### 5.1 An $\varepsilon$-net For Non-stationary Policies

In this section, we construct a set of polices which contains a near-optimal policy for any MDP. To define these policies, we first define a set of MDPs.

**Definition 5.1** (Discretized MDPs). *For given $\mathcal{S}$, $\mathcal{A}$, $H$, $s_1$ and $\varepsilon > 0$, define $\mathcal{M}_\varepsilon$ to be the set of MDPs $M = (\mathcal{S}, \mathcal{A}, P, R, H, s_1)$ such that*

- *Rewards are deterministic and for any $(s, a) \in \mathcal{S} \times \mathcal{A}$, $R(s, a) \in \{0, \varepsilon, 2\varepsilon, 3\varepsilon, \ldots, 1\}$;*
- *For each $(s, a, s') \in \mathcal{S} \times \mathcal{A} \times \mathcal{S}$, $P(s' \mid s, a) \in \{0, \varepsilon, 2\varepsilon, 3\varepsilon, \ldots, 1\}$;*

The following lemma gives an upper bound on the size of $\mathcal{M}_\varepsilon$. The proof is just by counting and we defer it to the supplementary material.

**Lemma 5.1.** $|\mathcal{M}_\varepsilon| \le (1/\varepsilon + 1)^{|\mathcal{S}|^2 |\mathcal{A}| + |\mathcal{S}||\mathcal{A}|}$.

Now we proceed to construct the $\varepsilon$-net for non-stationary policies, which is induced by $\mathcal{M}_\varepsilon$.

**Definition 5.2** ($\varepsilon$-net for Non-stationary Policies). *For given $\mathcal{S}$, $\mathcal{A}$, $H$ and $\varepsilon > 0$, define $\Pi_\varepsilon$ to be the set of polices such that*

$$\Pi_\varepsilon = \{\pi_M \mid \pi_M \text{ is an optimal policy for } M \in \mathcal{M}_\varepsilon\}.$$

*For each $M \in \mathcal{M}_\varepsilon$, when $M$ has multiple optimal policies, we add an arbitrary one to $\Pi_\varepsilon$.*

The following lemma shows $\Pi_\varepsilon$ is a valid $\varepsilon$-net in the sense that for any MDP, there exists a policy in $\Pi_\varepsilon$ which is near-optimal. The proof involves some perturbation analyses and we defer it to the supplementary material.

**Lemma 5.2.** *For any MDP $M = (\mathcal{S}, \mathcal{A}, P, R, H, s_1)$, there exists $\pi \in \Pi_\varepsilon$ such that $\pi$ is $8H|\mathcal{S}|\varepsilon$-optimal.*

We remark that since our final desired accuracy is $\varepsilon$, we will consider $\Pi_{\varepsilon/O(H|\mathcal{S}|)}$ which has size $(H|\mathcal{S}|/\varepsilon)^{O(|\mathcal{S}|^2|\mathcal{A}|)}$.

### 5.2 The Trajectory Simulator

In this section, we describe our algorithm for simulating trajectories. The algorithm is formally presented in Algorithm 1 and Algorithm 2. Algorithm 2 receives a parameter $\tau$ and uses a replay buffer $\mathcal{D}$ to store samples. Formally, $\mathcal{D} = \{\mathcal{D}_{s,a}\}_{s \in \mathcal{S}, a \in \mathcal{A}}$, where each $\mathcal{D}_{s,a}$ contains samples associated with state-action pair $(s, a)$, i.e.,

$$\mathcal{D}_{s,a} = \left[(s_{s,a}^{(1)}, r_{s,a}^{(1)}), (s_{s,a}^{(2)}, r_{s,a}^{(2)}), \ldots\right]$$

---

**Algorithm 1** SimAll

---

1: **Input:** failure probability $\delta_{\mathsf{sim}}$, policy set $\Pi$, number of trajectories $F$
2: $\tau \leftarrow 16|\mathcal{S}||\mathcal{A}|/\delta_{\mathsf{sim}} \cdot \log(4|\mathcal{S}||\mathcal{A}|/\delta_{\mathsf{sim}})$
3: **for** $i \in [F]$ **do**                                        ▷ Run $F$ copies of Algorithm 2 in parallel
4:     Set $\mathsf{SO}_i$ to be the $i$-th independent copy of $\mathtt{SimOne}(\tau)$ (Algorithm 2)
5: **for** $\pi \in \Pi$ **do**
6:     **for** $i \in [F]$ **do**
7:         $z_i^{\pi} \leftarrow \mathsf{SO}_i.\mathrm{SIMULATE}(\pi)$
8:     **if** $\sum_{i=1}^{F} \mathbb{I}[z_i^{\pi} \text{ is } \mathtt{Fail}] > 3\delta_{\mathsf{sim}}/2 \cdot F$ **then**
9:         **for** $i \in [F]$ **do**
10:             $z_i^{\pi} \leftarrow \mathsf{SO}_i.\mathrm{ROLLOUT}(\pi)$
11: **return** $\left\{z_i^{\pi}\right\}_{(i,\pi)\in[F]\times\Pi}$

---

and samples are sorted in chronological order. We also maintain $\Pi_{\mathcal{D}}$ in Algorithm 2 which is the set of policies used to generate $\mathcal{D}$. There are two subroutines in Algorithm 2. Subroutine SIMULATE takes an input policy $\pi$ and outputs either $\mathtt{Fail}$ or a trajectory for policy $\pi$. Subroutine ROLLOUT takes an input policy $\pi$, samples $\tau$ episodes for $\pi$ by interacting with the environment and stores all collected samples in the replay buffer $\mathcal{D}$. It also returns one of the $\tau$ trajectories sampled for for $\pi$. Moreover, whenever Subroutine ROLLOUT is invoked, samples in $\mathcal{D}$ are recollected so that independence among samples in the replay buffer $\mathcal{D}$ is ensured. Our analysis (Lemma B.2 in the supplementary material) critically relies on such independence, and therefore, it is unclear whether such recollecting step can be removed.

Algorithm 1 receives a failure probability $\delta_{\mathsf{sim}}$ and a policy set $\Pi$ as inputs. In Algorithm 1, we run $F$ independent copies of Algorithm 2 in parallel. For each policy $\pi$, for the $F$ independent copies of Algorithm 2, Algorithm 1 checks whether Subroutine SIMULATE returns $\mathtt{Fail}$ for too many times. If so, it calls Subroutine ROLLOUT for each copy of Algorithm 2 to collect samples and produce trajectories for $\pi$. Otherwise, it directly returns trajectories returned by Subroutine SIMULATE. Our main technical result is the following lemma which upper bounds the sample complexity of this procedure. The proof is provided in the supplementary material.

**Lemma 5.3.** *Suppose $F \geq 24/\delta_{\mathsf{sim}} \log(2|\Pi|/\delta_{\mathsf{overall}})$. With probability at least $1 - \delta_{\mathsf{overall}}/2$, Algorithm 1 at most interacts*

$$O\left(|\mathcal{S}||\mathcal{A}|/\delta_{\mathsf{sim}} \cdot \log(|\mathcal{S}||\mathcal{A}|/\delta_{\mathsf{sim}}) \cdot |\mathcal{S}|^2|\mathcal{A}|^2 \log^2 H \cdot F\right)$$

*episodes with the environment.*

### 5.3 Main Algorithm

In this section we present our final algorithm. The algorithm description is given in Algorithm 3. Our algorithm invokes Algorithm 1 on the set of policies defined in in Definition 5.2 to obtain trajectories for each policy, and simply returns the policy with largest empirical cumulative reward. The following lemma guarantees the correctness of our algorithm. The proof is deferred to the supplementary material.

**Lemma 5.4.** *With probability at least $1 - \delta_{\mathsf{overall}}/2$, Algorithm 3 returns an $\varepsilon$-optimal policy.*

Our main result, Theorem 4.1 is a direct implication of Lemma 5.3 and Lemma 5.4.

## 6 Discussion and Further Open Problems

This work provides an episodic, tabular reinforcement learning algorithm whose sample complexity only depends logarithmically with the planning horizon, thus resolving the open problem proposed in Jiang and Agarwal [2018]. This result is an exponential improvement on the dependency on $H$ over existing upper bounds. Furthermore, this works applies to a more general setting, where we only assume the total reward is bounded by one, without requiring any boundedness on instantaneous rewards (see Section 3).

---

**Algorithm 2** `SimOne`

---

1: **Input:** number of repetitions $\tau$
2: **function** SIMULATE($\pi$):
3:     **for** $(s,a) \in \mathcal{S} \times \mathcal{A}$ **do**
4:         Mark all elements in $\mathcal{D}_{s,a}$ as `unused`
5:     **for** $h \in \{1, 2, \ldots, H\}$ **do**
6:         **if** all elements in $\mathcal{D}_{s_h, \pi_h(s_h)}$ are marked as `used` **then**
7:             **return** Fail
8:         **else**
9:             Set $(s_{h+1}, r_h)$ to be the first element in $\mathcal{D}_{s_h, \pi_h(s_h)}$ that is marked as `unused`
10:            Mark $(s_{h+1}, r_h)$ (the first unused element in $\mathcal{D}_{s_h, \pi_h(s_h)}$) as `used`
11:    **return** $(s_1, \pi_1(s_1), r_1), (s_2, \pi_2(s_2), r_2), \ldots, (s_H, \pi_H(s_H), r_H)$
12: **function** ROLLOUT($\pi$)
13:    Set $\mathcal{D}_{s,a}$ to be an empty sequence for all $(s,a) \in \mathcal{S} \times \mathcal{A}$
14:    $\Pi_{\mathcal{D}} \leftarrow \Pi_{\mathcal{D}} \cup \{\pi\}$
15:    **for** $\pi' \in \Pi_{\mathcal{D}}$ **do**
16:        Sample $\tau$ trajectories for $\pi'$ by interacting with the environment
17:        Add all collected samples to $\mathcal{D}$
18:    **return** one of the $\tau$ trajectories sampled for $\pi$

---

**Algorithm 3** `Main`

---

1: **Input:** failure probability $\delta_{\mathsf{overall}}$, accuracy $\varepsilon$
2: Let $\Pi_{\varepsilon/(32H|\mathcal{S}|)}$ be the set of policies as defined in Definition 5.2
3: Invoke `SimAll` (Algorithm 1) with $\delta_{\mathsf{sim}} = \varepsilon/8$ and

$$F = \max\{64\log(4|\Pi_{\varepsilon/(32H|\mathcal{S}|)}|/\delta_{\mathsf{overall}})/\varepsilon^2, 192\log(2|\Pi_{\varepsilon/(32H|\mathcal{S}|)}|/\delta_{\mathsf{overall}})/\varepsilon\}$$

4: **for** each trajectory $z = (s_1, a_1, r_1), (s_2, a_2, r_2), \ldots, (s_H, a_H, r_H)$ returned by `SimAll` **do**
5:     Calculate $r(z) = \begin{cases} 0 & z \text{ is Fail} \\ \sum_{h=1}^{H} r_h & \text{otherwise} \end{cases}$
6: **for** $\pi \in \Pi_{\varepsilon/(32H|\mathcal{S}|)}$ **do**
7:     Calculate $\hat{r}(\pi) = \frac{1}{F} \sum_{i \in [F]} r(z_i^{\pi})$
8: **return** $\arg\max_{\pi \in \Pi_{\varepsilon/(32H|\mathcal{S}|)}} \hat{r}(\pi)$

---

**Conjectured Minimax Optimal Sample Complexity for Tabular RL.** Our upper bounds have a worse dependency on $|\mathcal{S}|, |\mathcal{A}|$ and $1/\varepsilon$ compared to existing results. One may conjecture that this suggests there is a tradeoff between obtaining a sublinear rate (in the model size) and obtaining a logarithmic dependence on $H$. However, our conjecture is that this is not the case, and that the PAC minimax-optimal sample complexity for episodic, tabular RL (to obtain an $\varepsilon$-optimal policy) is of the form:

$$\widetilde{O}\left(\frac{|\mathcal{S}||\mathcal{A}|\operatorname{poly}(\log H)}{\varepsilon^2}\right).$$

This conjecture, if true, would show that when sample complexity is measured by the number of episodes, then the difficulty of RL is a not a function of the horizon, and there is a sense in which RL is no more challenging than a contextual bandit problem.

We also conjecture a minimax optimal regret bound (rather than PAC) is of the same form.

**Computational Efficiency.** The computation complexity of our simulator scales as $|\Pi|$ due to that we need to simulate all policies in the set. The policy set we study in this paper has cardinality exponential in $|\mathcal{S}|$ and $|\mathcal{A}|$. We leave it as an open problem is to develop a polynomial time algorithm for the setting where $r_h \geq 0$ for $h \in [H]$ and $\sum_{h=1}^{H} r_h \leq 1$ whose sample complexity scales only logarithmically with $H$. One possible way is to exclude sub-optimal policies in $|\Pi|$ in our simulator without explicitly evaluating them.

It is also an important open question whether or not an upper confidence bound approach achieves the same polylogarithmic dependence in $H$. Such an algorithm would be promising for resolving the aforementioned conjectured minimax optimal rate.

**Generalization and Large State Space.**    While our paper shows, from sample complexity point of view, that long planning horizon is not an issue, in practice, the state space $\mathcal{S}$ can be huge. Recently, many works have established that with additional assumptions, e.g. low-rankness of the transition, functions approximations for $Q$-functions, etc, the sample complexity does not depend on $|\mathcal{S}|$ [Li et al., 2011, Wen and Van Roy, 2017, Krishnamurthy et al., 2016, Jiang et al., 2017, Dann et al., 2018, Du et al., 2019b, Feng et al., 2020, Du et al., 2019c, Zhong et al., 2019, Yang and Wang, 2019, Jin et al., 2019, Du et al., 2019a, Roy and Dong, 2019, Lattimore and Szepesvari, 2019, Du et al., 2020, Zanette et al., 2020].[7] However, to our knowledge, the sample complexity of all these work scales polynomially with $H$ with the only exceptions to require the transition being deterministic [Wen and Van Roy, 2017, Du et al., 2020]. We believe a fruitful direction is to develop algorithms for these settings with sample complexity that scales only logarithmically with $H$.

# Broader Impact

The focus of this paper is purely theoretical, and thus a broader impact discussion is not applicable.

# Disclosure of Funding

Ruosong Wang was supported in part by NSF IIS1763562, US Army W911NF1920104 and ONR Grant N000141812861. Part of the work is done while Simon S. Du was at the Institute for Advanced Study where he was supported by NSF grant DMS-1638352 and the Infosys Membership. Sham M. Kakade gratefully acknowledges funding from the ONR award N00014-18-1-2247, and NSF Awards CCF-1703574 and CCF-1740551.

## Footnotes

[2]Note that here we do not claim that our result strictly improves existing results. When $\varepsilon$ is sufficiently small (e.g. $\varepsilon \ll 1/H$), the sample complexity of our algorithm is worse than that of previous algorithms [Zanette and Brunskill, 2019]. In this paper, we primarily focus on the case when $\varepsilon$ is sufficiently large (e.g. $\varepsilon \gg 1/H$).

[3] $\widetilde{O}(\cdot)$ omits logarithmic factors.

[4] in [Dann and Brunskill, 2015], it appears that $\widetilde{O}(|\mathcal{S}|^2|\mathcal{A}|/\varepsilon^2)$ episodes are sufficient for $\varepsilon \in [0, 1/H]$. However, it is not clear whether the requirement of $\varepsilon \in [0, 1/H]$ can be relaxed to $\varepsilon \in [0, 1]$.

[5] See the motor control problem described in Section 2.1 in [Jiang and Agarwal, 2018].

[6] In fact, Zanette and Brunskill [2019] proved a even stronger result that the first term can be a problem-dependent quantity which is upper bounded by $|\mathcal{S}||\mathcal{A}|/\varepsilon^2$.

[7]Here we only focus on works where agent needs to explore the environment. There is another line of works that require a sufficiently good exploration policy, e.g., Fan et al. [2019].

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
