[Supplementary Material]

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

# A  Missing Proofs in Section 5.1

Throughout this section, without loss of generality, we assume $1/\varepsilon$ is a positive integer. In general, we may decrease $\varepsilon$ by a factor of at most two so that $1/\varepsilon$ is a positive integer.

The following definition is helpful in our analysis.

**Definition A.1.** *For an MDP $M = (\mathcal{S}, \mathcal{A}, P, R, H, \mu)$, we say a pair $(s, h) \in \mathcal{S} \times [H]$ is* admissible *with respect to $M$ if there exists a policy $\pi$ such that $\Pr[s_h = s \mid \pi] > 0$.*

Before presenting our analysis, we prove the following property regarding admissible pairs.

**Lemma A.1.** *For any admissible $(s, h) \in \mathcal{S} \times [H]$, for any $a \in \mathcal{A}$, the following hold:*

- $0 \le R(s, a) \le 1$ *almost surely;*

- $0 \le Q_h^\pi(s, a) \le 1$ *for any policy $\pi$;*

- $0 \le V_h^\pi(s) \le 1$ *for any policy $\pi$.*

*Proof of Lemma A.1.* Here we only prove $0 \le R(s, a) \le 1$. It can be similarly proved that $0 \le Q_h^\pi(s, a) \le 1$ and $0 \le V_h^\pi(s) \le 1$. Suppose $R(s, a) > 1$ or $R(s, a) < 0$ with non-zero probability. Since $(s, h)$ is admissible, there exists a policy $\pi$ such that $\Pr[s_h = s \mid \pi] > 0$. Consider the policy $\pi'$ defined to be:

$$\pi'_{h'}(s) = \begin{cases} \pi_{h'}(s) & h' < h \\ a & h' \ge h \end{cases}.$$

Clearly, $r_h > 1$ or $r_h < 0$ with non-zero probability, which violates the assumption that $\sum_{h=1}^{H} r_h \in [0, 1]$ and $r_h \ge 0$ for all $h \in [H]$ almost surely. $\square$

*Proof Lemma 5.1.* Since each $M \in \mathcal{M}_\varepsilon$ is uniquely defined by its $R$ and $P$, below we count the number of possible $R$ and $P$ respectively.

Since rewards are deterministic and for any $(s, a) \in \mathcal{S} \times \mathcal{A}$, $R(s, a) \in \{0, \varepsilon, 2\varepsilon, \ldots, 1\}$, there are $(1/\varepsilon + 1)^{|\mathcal{S}||\mathcal{A}|}$ different rewards in total.

Since for each $(s, a, s') \in \mathcal{S} \times \mathcal{A} \times \mathcal{S}$, $P(s' \mid s, a) \in \{0, \varepsilon, 2\varepsilon, \ldots, 1\}$, there are at most $(1/\varepsilon + 1)^{|\mathcal{S}|^2|\mathcal{A}|}$ different transitions in total.

Therefore, $|\mathcal{M}_\varepsilon| \le (1/\varepsilon + 1)^{|\mathcal{S}|^2|\mathcal{A}| + |\mathcal{S}||\mathcal{A}|}$. $\square$

*Proof of Lemma 5.2.* We first show that there exists $\hat{M} = \left(\mathcal{S}, \mathcal{A}, \hat{P}, \hat{R}, H, s_1\right) \in \mathcal{M}_\varepsilon$ such that the following hold:

- For any $(s, h) \in \mathcal{S} \times [H]$ admissible with respect to $M$, for any $a \in \mathcal{A}$, $|\hat{R}(s, a) - \mathbb{E}[R(s, a)]| \le \varepsilon$;

- For each $(s, a, s') \in \mathcal{S} \times \mathcal{A} \times \mathcal{S}$, $\left| P(s' \mid s, a) - \hat{P}(s' \mid s, a) \right| \le \varepsilon$;

- For each $(s, a, s') \in \mathcal{S} \times \mathcal{A} \times \mathcal{S}$, if $P(s' \mid s, a) = 0$ then $\hat{P}(s' \mid s, a) = 0$;

Below we construct such $\hat{P}$ and $\hat{R}$. By Lemma A.1 we have $\mathbb{E}[R(s, a)] \in [0, 1]$. Therefore, by setting $\hat{R}(s, a)$ to be closest real number in $\{0, \varepsilon, 2\varepsilon, \ldots, 1\}$, we have $|\hat{R}(s, a) - \mathbb{E}[R(s, a)]| \le \varepsilon$. Furthermore, for each $(s, a, s') \in \mathcal{S} \times \mathcal{A} \times \mathcal{S}$, we set

$$P'(s' \mid s, a) = \min\{p \in \{0, \varepsilon, 2\varepsilon, \ldots, 1\} \mid p \ge P(s' \mid s, a)\}.$$

Notice that $P'(s, a)$ may not always be a probability distribution. Clearly $P'(s' \mid s, a) \ge P(s' \mid s, a)$ for each $(s, a, s') \in \mathcal{S} \times \mathcal{A} \times \mathcal{S}$ and $\sum_{s' \in \mathcal{S}} P'(s' \mid s, a) = 1 + k\varepsilon$ for some positive integer $0 \le k \le |\mathcal{S}|$. Now for each $(s, a)$, we set $\hat{P}(s' \mid s, a) = P'(s' \mid s, a) - \varepsilon$ for an arbitrary $k$ states $s' \in \mathcal{S}$ with $P(s' \mid s, a) > 0$, and set $\hat{P}(s' \mid s, a) = P'(s' \mid s, a)$ for all other states $s'$. Clearly, $P'(s, a)$ is a probability distrbution for any $(s, a)$ and satisfies the desired property.

Now for any policy $\pi$, we use $V^\pi$ to denote the $V$-value of $\pi$ on MDP $M$, and use $\hat{V}^\pi$ to denote the $V$-value of $\pi$ on $\hat{M}$. $Q^\pi$ and $\hat{Q}^\pi$ are defined analogously. We prove that $|V^\pi - \hat{V}^\pi| \le 4|\mathcal{S}|H\varepsilon$ for any policy $\pi$ inductively by the following induction hypothesis:

- $|V_h^\pi(s) - \hat{V}_h^\pi(s)| \le (1 + (H-h)(|\mathcal{S}|+1))\varepsilon$ for any admissible $(s,h)$;

- $|Q_h^\pi(s,a) - \hat{Q}_h^\pi(s,a)| \le (1 + (H-h)(|\mathcal{S}|+1))\varepsilon$ for any admissible $(s,h)$ and any $a \in \mathcal{A}$.

When $h = H$, $V_H^\pi(s) = Q_H(s, \pi_H(s)) = \mathbb{E}[R(s, \pi_H(s))]$ and $\hat{V}_H^\pi(s) = \hat{Q}_H(s, \pi_H(s)) = \hat{R}(s, \pi_H(s))$. Therefore, the induction hypothesis holds when $h = H$ since $|\hat{R}(s,a) - \mathbb{E}[R(s,a)]| \le \varepsilon$.

Now we show the induction hypothesis holds for any $h < H$. For any $h < H$, consider any state $s$ such that $(s,h)$ is admissible. Notice that $V_h^\pi(s) = Q_h(s, \pi_h(s))$ and $\hat{V}_h^\pi(s) = \hat{Q}_h(s, \pi_h(s))$, and therefore $|V_h^\pi(s) - \hat{V}_h^\pi(s)| = |Q_h^\pi(s, \pi_h(s)) - \hat{Q}_h^\pi(s, \pi_h(s))|$. Furthermore,

$$Q_h^\pi(s,a) = \mathbb{E}[R(s,a)] + \sum_{s' \in \mathcal{S}} P(s' \mid s, a)V_{h+1}^\pi(s')$$

and

$$\hat{Q}_h^\pi(s,a) = \hat{R}(s,a) + \sum_{s' \in \mathcal{S}} \hat{P}(s' \mid s, a)V_{h+1}^\pi(s').$$

Therefore,

$$\left| Q_h^\pi(s,a) - \hat{Q}_h^\pi(s,a) \right|$$

$$\le \left| \mathbb{E}[R(s,a)] - \hat{R}(s,a) \right| + \sum_{s':P(s'|s,a)>0} \left| P(s' \mid s, a)V_{h+1}^\pi(s') - \hat{P}(s' \mid s, a)\hat{V}_{h+1}^\pi(s') \right|$$

$$\le \varepsilon + \sum_{s':P(s'|s,a)>0} \left( \left| P(s' \mid s, a) - \hat{P}(s' \mid s, a) \right| \cdot V_{h+1}^\pi(s') + \hat{P}(s' \mid s, a) \cdot \left| V_{h+1}^\pi(s') - \hat{V}_{h+1}^\pi(s') \right| \right)$$

$$\le (|\mathcal{S}|+1)\varepsilon + \sum_{s':P(s'|s,a)>0} \hat{P}(s' \mid s, a) \cdot \left| V_{h+1}^\pi(s') - \hat{V}_{h+1}^\pi(s') \right| \quad (V_{h+1}^\pi(s') \le 1 \text{ by Lemma A.1})$$

$$\le (|\mathcal{S}|+1)\varepsilon + (1 + (H - (h+1))(|\mathcal{S}|+1))\varepsilon$$
$$\qquad\qquad\qquad\qquad (\textstyle\sum_{s' \in \mathcal{S}} \hat{P}(s' \mid s, a) = 1 \text{ and induction hypothesis})$$

$$= (1 + (H-h)(|\mathcal{S}|+1))\varepsilon.$$

Thus, we have

$$\left| V_1^\pi(s_1) - \hat{V}_1^\pi(s_1) \right| \le 4|\mathcal{S}|H\varepsilon.$$

Finally, consider any optimal policy $\hat{\pi}$ of $\hat{M}$ and any optimal policy $\pi$ of $M$, we have

$$V_1^{\hat{\pi}}(s_1) \ge \hat{V}_1^{\hat{\pi}}(s_1) - 4|\mathcal{S}|H\varepsilon \ge \hat{V}_1^\pi(s_1) - 4|\mathcal{S}|H\varepsilon \ge V_1^\pi(s_1) - 8|\mathcal{S}|H\varepsilon.$$

Since $\hat{\pi} \in \Pi_\varepsilon$, the lemma holds. $\qquad\square$

## B  Missing Proofs in Section 5.2

In this section, we present the formal analysis of Algorithm 1 and Algorithm 2. Before we present our analysis, we first introduce some necessary notations.

**Definition B.1.** *For any policy $\pi$, for any state-action pair $(s,a) \in \mathcal{S} \times \mathcal{A}$, define $f^\pi(s,a) \in [H]$ to be a random variable defined as*

$$f^\pi(s,a) = \sum_{h=1}^{H} \mathbb{I}[(s,a) = (s_h, a_h) \mid \pi].$$

*I.e., $f^\pi(s,a)$ is the random variable which is the total number of times a trajectory induced by $\pi$ visits $(s,a)$.*

We additionally define the following quantity to characterize the number times a state-action pair is visited by a set of policies. Intuitively, given a success probability $\delta$, this quantity measures the maximum number of times a policy within a given policy set can visit a particular $(s, a)$ pair.

**Definition B.2.** *For a set of policies $\Pi$, for any $(s, a) \in \mathcal{S} \times \mathcal{A}$, define*

$$\mu_\delta^\Pi(s, a) = \max \left\{ \lambda \mid \lambda \in [0, H], \max_{\pi \in \Pi} \Pr[f^\pi(s, a) \geq \lambda] \geq \delta \right\}.$$

Note that $\mu_\delta^\Pi(s, a)$ is always a non-negative integer since for any state-action pair $(s, a) \in \mathcal{S} \times \mathcal{A}$, policy $\pi$ and real number $\lambda$,

$$\Pr\left[f^\pi(s, a) \geq \lambda\right] = \Pr\left[f^\pi(s, a) \geq \lceil \lambda \rceil\right].$$

Our next lemma states that for some policy $\pi$, if `SimOne` fails with high probability, then there exists a state-action pair that $\pi$ visits more frequently than all previous policies.

**Lemma B.1.** *For a policy $\pi \in \Pi$, suppose Subroutine SIMULATE in Algorithm 2 returns `Fail` with probability at least $\delta_{\mathsf{sim}}$ over the randomness of the generating process of the replay buffer $\mathcal{D}$. There exists $(s, a) \in \mathcal{S} \times \mathcal{A}$ such that*

$$\Pr\left[f^\pi(s, a) > \tau \cdot \frac{\delta_{\mathsf{sim}}}{4|\mathcal{S}|\,|\mathcal{A}|} \cdot \mu_{\delta_{\mathsf{sim}}/(2|\mathcal{S}||\mathcal{A}|)}^{\Pi_\mathcal{D}}(s, a)\right] \geq \frac{\delta_{\mathsf{sim}}}{2|\mathcal{S}|\,|\mathcal{A}|}$$

*where $\Pi_\mathcal{D}$ is the set of policies used to generate $\mathcal{D}$.*

*Proof of Lemma B.1.* Suppose for the sake of contradiction that for each $(s, a) \in \mathcal{S} \times \mathcal{A}$,

$$\Pr\left[f^\pi(s, a) > \tau \cdot \frac{\delta_{\mathsf{sim}}}{4|\mathcal{S}|\,|\mathcal{A}|} \cdot \mu_{\delta_{\mathsf{sim}}/(2|\mathcal{S}||\mathcal{A}|)}^{\Pi_\mathcal{D}}(s, a)\right] < \frac{\delta_{\mathsf{sim}}}{2|\mathcal{S}|\,|\mathcal{A}|}.$$

Let us denote

$$\Gamma(s, a) = \tau \cdot \frac{\delta_{\mathsf{sim}}}{4|\mathcal{S}|\,|\mathcal{A}|} \cdot \mu_{\delta_{\mathsf{sim}}/(2|\mathcal{S}||\mathcal{A}|)}^{\Pi_\mathcal{D}}(s, a).$$

For each $(s, a) \in \mathcal{S} \times \mathcal{A}$, we have

$$\Pr\left[f^\pi(s, a) > \Gamma(s, a)\right] < \frac{\delta_{\mathsf{sim}}}{2|\mathcal{S}|\,|\mathcal{A}|}.$$

Therefore, by a union bound over all state-action pairs $(s, a) \in \mathcal{S} \times \mathcal{A}$, with probability at least $1 - \delta_{\mathsf{sim}}/2$, for all $(s, a) \in \mathcal{S} \times \mathcal{A}$,

$$f^\pi(s, a) \leq \Gamma(s, a). \tag{1}$$

For each $(s, a) \in \mathcal{S} \times \mathcal{A}$, define $\mathcal{E}_{s,a}$ to be the event that

$$\mathcal{E}_{s,a} = \left\{ |\mathcal{D}_{s,a}| \geq \Gamma(s, a) \right\}.$$

By Definition B.2, there exists a policy $\pi_{s,a}^* \in \Pi_\mathcal{D}$ such that

$$\Pr\left[f^{\pi_{s,a}^*}(s, a) \geq \mu_{\delta_{\mathsf{sim}}/(2|\mathcal{S}||\mathcal{A}|)}^{\Pi_\mathcal{D}}(s, a)\right] \geq \delta_{\mathsf{sim}}/(2|\mathcal{S}|\,|\mathcal{A}|).$$

Now consider Line 16 in Subroutine ROLLOUT in Algortihm 2. Define

$$X_i = \begin{cases} 1 & \text{if } \sum_{h=1}^H \mathbb{I}[(s_h, a_h) = (s, a)] \geq \mu_{\delta_{\mathsf{sim}}/(2|\mathcal{S}||\mathcal{A}|)}^{\Pi_\mathcal{D}}(s, a) \text{ for the } i\text{-th trajectory of } \pi_{s,a}^* \\ 0 & \text{otherwise} \end{cases}.$$

Note $X_1, \ldots, X_\tau$ are i.i.d. random variables. By definition, $\mathbb{E}[X_i] \geq \delta_{\mathsf{sim}}/(2|\mathcal{S}|\,|\mathcal{A}|)$. Therefore, since $\tau = 16|\mathcal{S}|\,|\mathcal{A}|\,/\delta_{\mathsf{sim}} \cdot \log(4|\mathcal{S}|\,|\mathcal{A}|\,/\delta_{\mathsf{sim}})$, by Chernoff bound,

$$\Pr\left[\sum_{i=1}^\tau X_i \leq \tau \cdot \frac{\delta_{\mathsf{sim}}}{4|\mathcal{S}|\,|\mathcal{A}|}\right] \leq \exp\left(-\frac{\tau \delta_{\mathsf{sim}}/(2|\mathcal{S}|\,|\mathcal{A}|)}{8}\right) \leq \frac{\delta_{\mathsf{sim}}}{4|\mathcal{S}|\,|\mathcal{A}|}.$$

Therefore,

$$\Pr[\mathcal{E}_{s,a}] \geq \Pr\left[\sum_{i=1}^{\tau} X_i \geq \tau \cdot \frac{\delta_{\mathsf{sim}}}{4|\mathcal{S}|\,|\mathcal{A}|}\right] \geq 1 - \frac{\delta_{\mathsf{sim}}}{4|\mathcal{S}|\,|\mathcal{A}|}.$$

It follows that with probability at least $1 - \delta_{\mathsf{sim}}/4$, for all $(s,a) \in \mathcal{S} \times \mathcal{A}$,

$$|\mathcal{D}_{s,a}| \geq \Gamma(s,a). \tag{2}$$

By a union bound over (1) and (2), with probability at least $1 - \frac{3\delta_{\mathsf{sim}}}{4}$, for all $(s,a) \in \mathcal{S} \times \mathcal{A}$,

$$|\mathcal{D}_{s,a}| \geq \Gamma(s,a) \geq f^{\pi}(s,a),$$

in which case Subroutine SIMULATE does not return Fail. This contradicts the assumption that Subroutine SIMULATE returns Fail with probability at least $\delta_{\mathsf{sim}}$. $\square$

Now we discuss the implication of Lemma B.1. Note that Algorithm 2 interacts with the environment to sample trajectories only when Subroutine SIMULATE fails with probability at least $\delta_{\mathsf{sim}}$. By Lemma B.1, when Algorithm 2 interacts with the environment to sample trajectories, $\mu^{\Pi_{\mathcal{D}}}_{\delta_{\mathsf{sim}}/(2|\mathcal{S}||\mathcal{A}|)}(s,a)$ doubles or changes from 0 to 1 for some $(s,a) \in \mathcal{S} \times \mathcal{A}$ since $\tau \delta_{\mathsf{sim}}/(4|\mathcal{S}|\,|\mathcal{A}|) > 2$. However, $\mu^{\Pi_{\mathcal{D}}}_{\delta_{\mathsf{sim}}/(2|\mathcal{S}||\mathcal{A}|)}(s,a)$ is always upper bounded by $H$. Therefore, the total number of calls to Subroutine ROLLOUT in Algorithm 1 is upper bounded by $O(|\mathcal{S}||\mathcal{A}|\log H)$. Our next lemma guarantees that whenever Algorithm 1 invokes Subroutine ROLLOUT, the probability that Subroutine SIMULATE returns Fail is at least $\delta_{\mathsf{sim}}$, and when Subroutine ROLLOUT is not invoked, the probability that Subroutine SIMULATE returns Fail is at most $2\delta_{\mathsf{sim}}$.

**Lemma B.2.** *Suppose $F \geq 24/\delta_{\mathsf{sim}} \cdot \log(2|\Pi|/\delta_{\mathsf{sim}})$ in Algorithm 1. With probability at least $1 - \delta_{\mathsf{sim}}/(2|\Pi|)$, each time Line 8 in Algorithm 1 is executed, the following hold:*

- *when $\sum_{i=1}^{F} \mathbb{I}[z_i^{\pi} \text{ is } \mathtt{Fail}] > 3\delta_{\mathsf{sim}}/2 \cdot F$, the probability that Subroutine SIMULATE returns Fail is at least $\delta_{\mathsf{sim}}$ over the randomness of the generating process of the replay buffer $\mathcal{D}$;*

- *when $\sum_{i=1}^{F} \mathbb{I}[z_i^{\pi} \text{ is } \mathtt{Fail}] \leq 3\delta_{\mathsf{sim}}/2 \cdot F$, the probability that Subroutine SIMULATE returns Fail is at most $2\delta_{\mathsf{sim}}$ over the randomness of the generating process of the replay buffer $\mathcal{D}$.*

*Proof of Lemma B.2.* Let $Y_i = \mathbb{I}[z_i^{\pi} \text{ is } \mathtt{Fail}]$. Note that each time Subroutine ROLLOUT is invoked, all samples in $\mathcal{D}$ are recollected. Therefore, for any given time step of the algorithm, $\{Y_i\}_{i=1}^{F}$ are independent random variables.

If $\Pr[Y_i = 1] < \delta_{\mathsf{sim}}$, by Chernoff bound,

$$\Pr\left[\sum_{i=1}^{F} Y_i \geq 3\delta_{\mathsf{sim}}/2 \cdot F\right] \leq \exp(-\delta_{\mathsf{sim}} F/24) \leq \frac{\delta_{\mathsf{overall}}}{2|\Pi|}.$$

On the other hand, if $\Pr[Y_i = 1] \geq 2\delta_{\mathsf{sim}}$, by Chernoff bound,

$$\Pr\left[\sum_{i=1}^{F} Y_i \leq 3\delta_{\mathsf{sim}}/2 \cdot F\right] \leq \exp(-\delta_{\mathsf{sim}} F/16) \leq \frac{\delta_{\mathsf{overall}}}{2|\Pi|}.$$

Thus the lemma holds. $\square$

**Lemma B.3.** *Suppose $F \geq 24/\delta_{\mathsf{sim}} \cdot \log(2|\Pi|/\delta_{\mathsf{overall}})$ in Algorithm 1. Let $\Pi_{\mathcal{D}}$ be the set of policies maintained by Algorithm 2 before executing Line 14, and let $\hat{\Pi}_{\mathcal{D}}$ be the set of policies maintained after executing Line 14, i.e., $\hat{\Pi}_{\mathcal{D}} = \Pi_{\mathcal{D}} \cup \{\pi\}$. With probability at least $1 - \delta_{\mathsf{overall}}/(2|\Pi|)$, there exists $(s,a) \in \mathcal{S} \times \mathcal{A}$, such that*

$$\mu^{\hat{\Pi}_{\mathcal{D}}}_{\delta_{\mathsf{sim}}/(2|\mathcal{S}||\mathcal{A}|)}(s,a) \geq \max\left(2 \cdot \mu^{\Pi_{\mathcal{D}}}_{\delta_{\mathsf{sim}}/(2|\mathcal{S}||\mathcal{A}|)}(s,a), 1\right).$$

*Proof of Lemma B.3.* By Lemma B.2, with probability at least $1 - \delta_{\text{overall}}/(2|\Pi|)$, for the added policy $\pi$, the probability that Subroutine SIMULATE returns Fail is at least $\delta_{\text{sim}}$. By Lemma B.1, there exists $(s, a) \in \mathcal{S} \times \mathcal{A}$ such that

$$\Pr\left[ f^\pi(s, a) > \tau \cdot \frac{\delta_{\text{sim}}}{4|\mathcal{S}||\mathcal{A}|} \cdot \mu_{\delta_{\text{sim}}/(2|\mathcal{S}||\mathcal{A}|)}^{\Pi_\mathcal{D}}(s, a) \right] \geq \frac{\delta_{\text{sim}}}{2|\mathcal{S}||\mathcal{A}|}.$$

If $\mu_{\delta_{\text{sim}}/(2|\mathcal{S}||\mathcal{A}|)}^{\Pi_\mathcal{D}}(s, a) = 0$, we have

$$\Pr[f^\pi(s, a) > 0] = \Pr[f^\pi(s, a) \geq 1] \geq \frac{\delta_{\text{sim}}}{2|\mathcal{S}||\mathcal{A}|}.$$

Otherwise, we have

$$\Pr\left[ f^\pi(s, a) \geq 2 \cdot \mu_{\delta_{\text{sim}}/(2|\mathcal{S}||\mathcal{A}|)}^{\Pi_\mathcal{D}}(s, a) \right] \geq \frac{\delta_{\text{sim}}}{2|\mathcal{S}||\mathcal{A}|}.$$

$\square$

*Proof of Lemma 5.3.* Notice that our algorithm interacts with the environment only when Subroutine ROLLOUT in Algorithm 2 is invoked. By Lemma B.3 and union bound, with probability at least $1 - \delta_{\text{overall}}/2$, whenever Subroutine ROLLOUT is invoked, there exists $(s, a) \in \mathcal{S} \times \mathcal{A}$ such that $\mu_{\delta_{\text{sim}}/(2|\mathcal{S}||\mathcal{A}|)}^{\Pi_\mathcal{D}}(s, a)$ is increased from 0 to 1, or $\mu_{\delta_{\text{sim}}/(2|\mathcal{S}||\mathcal{A}|)}^{\Pi_\mathcal{D}}(s, a)$ is increased by a factor of 2. Since $\mu_{\delta_{\text{sim}}/(2|\mathcal{S}||\mathcal{A}|)}^{\Pi_\mathcal{D}}(s, a) \leq H$, with probability at least $1 - \delta_{\text{sim}}/2$, Subroutine ROLLOUT is invoked for at most $O(|\mathcal{S}||\mathcal{A}| \log H)$ times. Hence $|\Pi_\mathcal{D}| = O(|\mathcal{S}||\mathcal{A}| \log H)$. Finally, whenever Subroutine ROLLOUT is invoked, the algorithm samples at most $F|\Pi_\mathcal{D}|\tau$ trajectories by interacting with the environment. Therefore, with probability at least $1 - \delta_{\text{sim}}/2$, the total number of trajectories sampled by the algorithm is upper bounded by $O(F\tau \cdot (|\mathcal{S}||\mathcal{A}| \log H)^2)$. $\square$

## C  Missing Proofs in Section 5.3

**Lemma C.1.** *For each policy $\pi \in \Pi_{\varepsilon/(32H|\mathcal{S}|)}$, for the value $\hat{r}(\pi)$ calculated in Line 7 of Algorithm 3, with probability at least $1 - \delta_{\text{overall}}/(2|\Pi_{\varepsilon/(32H|\mathcal{S}|)}|)$,*

$$\left| \hat{r}(\pi) - \mathbb{E}\left[ \sum_{h=1}^{H} r_h \mid \pi \right] \right| \leq 5\varepsilon/16.$$

*Proof of Lemma C.1.* For those policies $\pi \in \Pi_\mathcal{D}$, notice that $\{z_i^\pi\}_{i \in [F]}$ are sampled by interacting with the environment. Since all reward values are positive and cumulative reward is upper bounded by 1 almost surely, by Chernoff bound,

$$\Pr\left[ \left| \hat{r}(\pi) - \mathbb{E}\left[ \sum_{h=1}^{H} r_h \mid \pi \right] \right| \leq \varepsilon/8 \right] \geq 1 - 2\exp(-F\varepsilon^2/64) \geq 1 - \delta_{\text{overall}}/(2|\Pi_{\varepsilon/(32H|\mathcal{S}|)}|).$$

For those policies $\pi \notin \Pi_\mathcal{D}$, notice that $\{z_i^\pi\}_{i \in [F]}$ have the same distribution as $F$ independent trajectories sampled by interacting with the environment, except that at most $3\delta_{\text{sim}}/2 \cdot F = 3\varepsilon/16 \cdot F$ trajectories are replaced with Fail. If all trajectories are independently sampled by interacting with the environment, by Chernoff bound, with probability at least $1 - \delta_{\text{overall}}/(2|\Pi_{\varepsilon/(32H|\mathcal{S}|)}|)$,

$$\left| \hat{r}(\pi) - \mathbb{E}\left[ \sum_{h=1}^{H} r_h \mid \pi \right] \right| \leq \varepsilon/8.$$

Since cumulative reward is in $[0, 1]$ almost surely, by replacing at most $3\varepsilon/16 \cdot F$ trajectories with Fail, $\hat{r}(\pi)$ is changed by at most $3\varepsilon/16$. Therefore, with probability at least $1 - \delta_{\text{overall}}/(2|\Pi_{\varepsilon/(32H|\mathcal{S}|)}|)$,

$$\left| \hat{r}(\pi) - \mathbb{E}\left[ \sum_{h=1}^{H} r_h \mid \pi \right] \right| \leq 5\varepsilon/16.$$

$\square$

*Proof of Lemma 5.4.* By Lemma 5.2, there exists a $\varepsilon/4$-optimal policy $\pi' \in \Pi_{\varepsilon/(32H|\mathcal{S}|)}$. By Lemma C.1 and a union bound over $\Pi$, with probability at least $1 - \delta_{\text{overall}}/2$, for all policy $\pi \in \Pi_{\varepsilon/(32H|\mathcal{S}|)}$,

$$\left| \hat{r}(\pi) - \mathbb{E}\left[ \sum_{h=1}^{H} r_h \mid \pi \right] \right| \leq 5\varepsilon/16.$$

Let $\pi$ be the policy returned by algorithm. Conditioned on the event mentioned above, we have

$$\mathbb{E}\left[ \sum_{h=1}^{H} r_h \mid \pi \right] \geq \hat{r}(\pi) - 5\varepsilon/16 \geq \hat{r}(\pi') - 5\varepsilon/16 \geq \mathbb{E}\left[ \sum_{h=1}^{H} r_h \mid \pi' \right] - 5\varepsilon/8 \geq \mathbb{E}\left[ \sum_{h=1}^{H} r_h \mid \pi^* \right] - \varepsilon.$$

$\square$