[Reviews · NeurIPS 2020]

Review 1

Summary and Contributions: This paper studies the episodic MDP under a slightly more general assumption on the reward function (assumption 3.2) and compares the sample complexity in the number of episodes instead of the number of timestep. Previous results all have poly in H dependence in the number of episodes, and this paper is the first one that proposes an algorithm to achieve polylog H dependence on the number of episodes under this general assumption. Hence, this paper provides a negative answer to the COLT 2018 open problem by Jiang and Agarwal. This paper also proposes the epsilon-net concept and utilizes online trajectory synthesis, both of which are interesting themselves. I also believe this paper will motivate further future research questions.

Strengths: The idea of epsilon net is novel and new to me. This paper uses the online trajectory synthesis which only has low sample complexity. This paper proposes the first algorithm that has polylog H dependence on the number of episodes under more general assumption on reward functions, and therefore solves a COLT 2018 open problem.

Weaknesses: The proposed algorithm is not computationally efficient. The result seems to have a lot of room to improve, w.r.t. |S|, |A|, and 1/epsilon.

Correctness: I checked the proof in detail and the proof looks rigorous to me.

Clarity: This paper is clearly written and easy to follow. The main idea and concept are conveyed in the main text and the proof is deferred to the appendix. I think this paper is generally easy to read for the RL community.

Relation to Prior Work: The comparison between this work and the previous tabular PAC result are discussed. The difference in the assumptions and when and why this more general setting is interesting is well illustrated. Some related work about the trajectory synthesis method is also mentioned in the paper.

Reproducibility: Yes

Additional Feedback: After rebuttal: I read the author response and other reviews, and would like to keep my original score. After the rebuttal, it's still unclear why the authors would like to clear the dataset and hope to see more discussions on that in a later version. Besides, as other reviewer pointed out, it would be clearer if the authors could discuss the regime (epsilon >> 1/H) that they advance the prior result in the intro or abstract. Beyond that I think achieving polylog(H) result in the setting epsilon >> 1/H is still interesting and the analysis is novel. Regarding the epsilon-net, I think the idea of discretizing MDP is new to me. However, it's not surprising to see the result thereafter. So the authors may consider moving some of those parts to the appendix. ================================================================================================================================================================================================================================================ Overall, I think this paper has enough contribution for the conference. The result is solid and will motivate further research questions (e.g. generalization to the rich observation setting, the improvement on the computation side, and deeper understanding about the dependence on the horizon) to the community. The paper is easy to follow, and the main concepts and algorithms are well exposed. As mentioned above, this paper studies the episodic MDP under a slightly more general assumption on the reward function (assumption 3.2). This paper compares the sample complexity in the number of episodes instead of the number of timestep. This would be a fairer comparison for the long horizon problem since we expect the long horizon problem would pay more timestep due to larger H, while comparing the number of episodes can avoid such issue. By proving a bound that only has polylog dependence on H, this paper solves the COLT 2018 open problem by Jiang and Agrawal. The main idea of the algorithm is to first construct a set of MDPs by discretizing the transition function and the reward function, and further obtain a set of candidate policies (all the optimal policies for these different MDPs). Then the algorithm collects a dataset that is possible to approximate evaluate all these candidate policies. Finally, the algorithm evaluates all these policies and pick the best one. Since the true MDP is close to at least one of these MDPs (say M0), the optimal policy under M0 (say pi_0) must be near optimal. In addition, the algorithm is guaranteed to pick one that has close performance as pi_0, thus eventually gets a near-optimal policy. The main technical question is how to construct such dataset. A naive way to interact with the environment to sample according to all candidate policies will immediately fail. The authors design a clever way to only sample when necessary and use a potential function to analyze the number of samples. I only have some minor questions as below. 1. In line 13 of the algorithm, the dataset is cleared. I think the reason is to get independence in the analysis, however, it seems a little bit inefficient. Is it possible to avoid that and maybe with different analysis? 2. The authors mentioned on line 294-297 that some previous algorithm can avoid the poly H dependence in deterministic environment. Can authors comment why that would happen or maybe some intuition?


Review 2

Summary and Contributions: This paper refutes the conjecture that poly(H) lower bound is needed for episodic RL, while the existing lower bounds don't depend on H. In this paper, they show that the sample complexity depending only on poly(log(H)) is possible by designing a sampling scheme to reduce the sample complexity and sacrificing computation complexity. In general speaking, this paper contributes a new approach to resolve the open problem in COLT 2018 for episodic MDP, which surprisingly breaks our intuition that long-horizon MDP is more difficult than short-horizon MDP.

Strengths: I have checked the proof of theoretical results and the results and solid. I reckon that this paper breaks the traditional intuition that long-horizon MDP is harder than the short-horizon MDP, though the computation complexity scales exponentially in state space and action space. The most interesting idea is reusing exploration samples to reduce sample complexity instead of solving dynamic programming backward as prior work does. I reckon this paper provides a new perspective for RL communities to design new efficient learning algorithms.

Weaknesses: I understand this paper solves the problem with some unwanted sacrifice(computation efficiency). In terms of solving sample complexity itself, I reckon this paper does well. There are two aspects in terms of computation inefficiency. 1. The number of exploration/simulation policies is exponential in state/action space. 2. For each policy, the parallel number F is poly(SA) and there is no sublinear acceleration(indeed linear in F) in sample complexity.

Correctness: 1. Line 477/478 in Supplementary, I guess 'ROLLOUT' means 'SIMULATE' 2. Algorithm 3, Line3: SimAll is Algorithm1 not 2 Besides, I have one question about this paper. This paper assumes the transition probability is independent of horizons (P_h are all the same for h=1,...,H). I'm wondering whether the sample complexity still depends on poly(log H) if transition probability varies as horzion changes (P_h are not the same for h=1,...,H).

Clarity: yes

Relation to Prior Work: yes

Reproducibility: Yes

Additional Feedback: Refer to weakness for details ==================== I have read the authors' response and I decide to keep my score


Review 3

Summary and Contributions: The authors propose an algorithm for PAC learning in tabular finite horizon MDPs and present PAC bounds. Their work is motivated by a recent conjecture from Jian and Agarwal 2018 which asked whether a lower bound is polynomial in H and in particular (SAH / epsilon^2) would be the right scaling. In this paper the authors provide an algorithm that has a PAC bound that scales as (S^5 A^4/epsilon^3) with only a log dependence on H.

Strengths: The algorithm itself is quite interesting. It involves simulating each policy using prior samples and only running in the real environment if it the number of times it was not possible to use prior data to create a simulated trajectory exceeds a certain threshold. This work helps bring us closer to answering the hardness of acting in long horizon vs short horizon settings.

Weaknesses: The approach and results are interesting but the resulting conclusion overclaims. In particular, in terms of H and epsilon, these results seem stronger than prior results of Zanette and Brunskill 2019 in the regime where 1/epsilon < H and worse when those other results 1/epsilon > H. When 1/epsilon ~= H, their result would yield SAH/ epsilon^2 vs this prior work would have SA / epsilon^2 and SAH / epsilon, and therefore be strictly worse. This is ignoring their dependence on S and A (which is roughly S^5 A^4) which is substantially worse than prior results (which are S*A). This isn’t to say this result is not valuable, but it adds benefit under particular assumptions of the relation to H and epsilon, and it’s important to clarify these differences which also make the landscape of contributions more nuanced. The computational complexity is exponential in the S & A space The key aspects of the algorithm that enable this result would be very helpful to highlight in the main paper. For example, defining the failure probability (and how this changes over time as more policies are evaluated) is critical to the resulting bounds, but this is not explored in the main text. Similarly, it would be great to see more details in the main text about why a new set of trajectories from all prior policies are needed every time the rollout is executed. Section 5.1 could be kept in the appendix since it’s fairly straightforward to see why this many MDPs would enable a near optimal policy for any possible MDP, and the novelty lies more in these other aspects.

Correctness: I have not went through the full proofs in the appendix. --- Post rebuttal and reading others' reviews and discussion Thanks for your feedback. I think your proposed changes will strengthen the work including your promise to explicitly discuss the regime of epsilon under which your work improves over prior work. I agree that it's useful to have work that is polynomial in the S, A, and log in H and this makes a helpful contribution.

Clarity: See weaknesses.

Relation to Prior Work: Somewhat. I think a key limitation of the work is that upon a quick read it seems to suggest that it definitively answers a prior conjecture, but the situation is more nuanced. It outperforms in a particular regime of epsilon and H whereas some prior work outperforms it in other regimes.

Reproducibility: No

Additional Feedback: Ultimately the direction is interesting but the contribution is more limited than the introduction suggests, providing value over a different regime of epsilon and H than recent related results, and the algorithm is not computationally tractable.

[Author Response · NeurIPS 2020]

We thank all the reviewers for their valuable feedback and appreciating our contributions. Please find our response to
each individual reviewer below.

—— **To Reviewer #2** ——

**Line 13 of the algorithm.**   As correctly observed by Reviewer #1, in Line 13 of Algorithm 2, we clear the dataset and
retake samples to guarantee the independence of samples in the dataset. Our current analysis (Lemma B.2) critically
relies on such independence, and it is unclear if the algorithm is still correct after removing this step. We will add more
discussion on this step in the next version.

**Avoid $\text{poly}(H)$ dependence in deterministic environment.**   In deterministic systems, if a state-action pair $(s, a)$ is
reachable, then there exists a policy that always visits $(s, a)$, and in order to find the optimal policy, it suffices for the
agent to visit each reachable state-action pair once. Therefore, a simple algorithm would be the following: whenever
there exists a state-action pair $(s, a)$ that has not been visited, sample a trajectory to try to visit $(s, a)$, observe the
reward value $r$ and the transition $s'$, and mark $(s', a')$ to be unvisited for all actions $a'$. Clearly, after sampling $|\mathcal{S}| \times |\mathcal{A}|$
trajectories, the agent should have visited all reachable state-action pairs, at which point the agent could output the
optimal policy by planning on the learned model.

—— **To Reviewer #3** ——

**What if transition probability varies as horizon changes.**   If the transition operator varies as the horizon changes,
our algorithm can no longer achieve $\text{poly}(\log H)$ dependency. To name one reason, the size of the $\varepsilon$-net defined in
Section 5.1 now has exponential dependency on $H$. However, in such a setting, one can prove a lower bound of $\Omega(H)$
on the number of episodes. Such a lower bound can be proved by, e.g., using the standard combination lock environment.
We will add more discussion on this in the next version.

**Typos.**   Thanks for pointing out. We will fix these typos in the next version.

—— **To Reviewer #4** ——

**Comparison with previous results.**   First of all, we do not claim our result strictly improves existing results. The
current discussions on previous results in Section 3 primarily focus on their dependency on $H$. We totally agree that
when $\varepsilon$ is sufficiently small (e.g. $\varepsilon \ll 1/H$), the sample complexity of our algorithm is worse than that of previous
algorithms. In the next version, we will make this point explicit in the introduction, and provide more comparisons
between the sample complexity of our algorithm and that of previous algorithms to make the complexity landscape
clearer. Meanwhile, as mentioned by Reviewer #2, this paper is the first one that proposes an algorithm to achieve
$\text{poly}(\log H)$ dependence on the number of episodes.

**Answering the open problem in [Jiang and Agarwal, 2018].**   In terms of answering the open problem in [Jiang
and Agarwal, 2018], their problem statement is *"Can we prove a lower bound that depends polynomially in $H$?"*
Furthermore, [Jiang and Agarwal, 2018] mentioned explicitly that the relevant setting is when $\varepsilon \gg 1/H$ (see the
paragraph before Section 2.2 in [Jiang and Agarwal, 2018]). In this sense, our work resolves the open problem in [Jiang
and Agarwal, 2018] with a negative answer.

**The key aspects of the algorithm that enable this result would be very helpful to highlight in the main paper.**
Right now we have provided a technical overview in Section 4.1. We are happy to add more discussion on the novel
aspects of the analysis in the next version. Thanks for the suggestion!

**Why a new set of trajectories are needed every time the rollout is executed.**   As correctly observed by Reviewer
#2, in Line 13 of Algorithm 2, we clear the dataset and retake samples to guarantee the independence of samples in the
dataset. See the discussion above regarding this for more details.

**Section 5.1 could be kept in the appendix.**   We decided to keep Section 5.1 in the main text since it could be of
interest to a broad audience. For example, Reviewer #2 mentioned that "the idea of epsilon net is novel and new to me".
However, we are glad to move parts of Section 5.1 to the appendix and add more discussion on other novel aspects of
our analysis.

[Meta-Review · NeurIPS 2020]

This paper addresses a COLT 2018 open problem regarding whether there exists an algorithm for tabular, episodic ("long horizon") RL whose sample complexity is polynomial in the length of the horizon, H. It answers in the affirmative, providing an algorithm and PAC bound that has a poly-log dependence on H. This result furthers our understanding of episodic RL and shows that it is indeed not much more difficult than "short horizon" RL (with a few assumptions). The reviewers agree that this work is interesting and timely, and that the paper is more or less clear and well written. However, the reviewers also agree that the paper might be overstating its contributions a bit. The proposed bounds would be stronger than prior work in the setting where H > 1/epsilon, but worse when this does not hold, due to some pretty hefty polynomial terms involving the state and action space sizes, S and A. For the sake of honesty, the paper should be upfront about this limitation. Another critique is that the computational complexity of the proposed algorithm is bad. But this is somewhat expected of theoretical results, and does not detract from fact that there exists an algorithm that settles the open problem. It may be that more efficient algorithms have the same sample complexity; this would make nice followup work. I strongly encourage the authors to incorporate *all* of the feedback from the reviews -- especially w.r.t. over-claiming -- when finalizing the paper.